# Reduced High-Density Lipoprotein Cholesterol Is an Independent Determinant of Altered Bone Quality in Women with Type 2 Diabetes

**DOI:** 10.3390/ijms24076474

**Published:** 2023-03-30

**Authors:** Sara Dule, Ilaria Barchetta, Flavia Agata Cimini, Giulia Passarella, Arianna Dellanno, Tiziana Filardi, Vittorio Venditti, Enrico Bleve, Diego Bailetti, Elisabetta Romagnoli, Susanna Morano, Marco Giorgio Baroni, Maria Gisella Cavallo

**Affiliations:** 1Department of Experimental Medicine, Sapienza University, 00161 Rome, Italy; 2Department of Clinical Medicine, Public Health, Life and Environmental Sciences (MeSVA), University of L’Aquila, 67100 L’Aquila, Italy; 3Neuroendocrinology and Metabolic Diseases, IRCCS Neuromed, 86077 Pozzilli, Italy

**Keywords:** osteoporosis, osteopenia, fracture risk, lipid metabolism, metabolic syndrome, insulin resistance, trabecular bone score

## Abstract

Type 2 diabetes mellitus (T2DM) is associated with an increased fracture risk. Our study aimed to explore differences in bone alterations between T2DM women and controls and to assess clinical predictors of bone impairment in T2DM. For this observational case control study, we recruited 126 T2DM female patients and 117 non-diabetic, age- and BMI-comparable women, who underwent clinical examination, routine biochemistry and dual-energy X-ray absorptiometry (DXA) scans for bone mineral density (BMD) and trabecular bone score (TBS) assessment-derived indexes. These were correlated to metabolic parameters, such as glycemic control and lipid profile, by bivariate analyses, and significant variables were entered in multivariate adjusted models to detect independent determinants of altered bone status in diabetes. The T2DM patients were less represented in the normal bone category compared with controls (5% vs. 12%; *p* = 0.04); T2DM was associated with low TBS (OR: 2.47, C.I. 95%: 1.19–5.16, *p* = 0.016) in a regression model adjusted for age, menopausal status and BMI. In women with T2DM, TBS directly correlated with plasma high-density lipoprotein cholesterol (HDL-c) (*p* = 0.029) and vitamin D (*p* = 0.017) levels. An inverse association was observed with menopausal status (*p* < 0.001), metabolic syndrome (*p* = 0.014), BMI (*p* = 0.005), and waist circumference (*p* < 0.001). In the multivariate regression analysis, lower HDL-c represented the main predictor of altered bone quality in T2DM, regardless of age, menopausal status, BMI, waist circumference, statin treatment, physical activity, and vitamin D (*p* = 0.029; R^2^ = 0.47), which likely underlies common pathways between metabolic disease and bone health in diabetes.

## 1. Introduction

Type 2 diabetes mellitus (T2DM) and osteoporosis are chronic conditions with increasing prevalence worldwide, in part due to aging populations. In 2021, the global T2DM prevalence was estimated to be 10.5% and is expected to rise to 12.2% in 2045 [1], causing life threatening, disabling, costly complications, and reducing life expectancy [2]. On the other hand, the prevalence of osteoporosis in the world is reported to be 18.3%, increasing to 23.1% in females [3]. In Italy, only 8.1% of the population report having osteoporosis, and the prevalence progressively increases with advancing age, particularly in females, affecting 47% of females aged > 74 years [4].

Over the last years, a considerable amount of data has indicated that the metabolic and endocrine alterations of diabetes affect bone health. The exact mechanisms leading to bone structure alterations in T2DM are still not fully understood [5]. Different pathways have been proposed in the recent years: inhibition of osteoblastogenesis [6] and alteration of osteoblast’s gene expression by hyperglycaemia [7], advanced glycation end products (AGEs) cross-linking with collagen fibers [8], and increased levels of pro-inflammatory cytokines [9].

The coexistence of T2DM and osteoporosis determines poorer health outcomes [10,11,12,13,14]. Patients with T2DM have a 19% increased risk of any fragility fracture [10], independent of the frequency of falls and despite having higher BMD [11,12]. Diabetic patients also have an increased risk of impaired fracture healing [13] and higher post-fracture mortality [14] when compared with non-diabetic patients.

In the absence of fragility fractures, osteoporosis is usually diagnosed through BMD measurement by dual-energy X-ray absorptiometry (DXA) [15]. However, in patients with T2DM, BMD is often normal to increased [16], making it an unreliable parameter to assess fracture risk. Bone fragility in diabetes is dependent not only on bone mineralization, but also on bone structure alterations. Data indicate the presence of cortical bone alterations in patients with T2DM [17], but the effects on trabecular bone remain controversial. In fact, studies conducted on bone biopsies in T2DM have shown the presence of fewer and thinner trabeculae leading to increased bone fragility, both in rats [18] and in humans [19]. Conversely, bone trabecular microarchitecture was shown to be preserved or even improved in diabetic patients when explored with other imaging techniques, such as High-Resolution Peripheral Quantitative Computed Tomography (HR-pQCT) [17,20]. The Trabecular Bone Score (TBS) is a BMD-independent parameter of the spine DXA image which evaluates bone texture and is, therefore, an indirect estimator of the trabecular bone quality. Evidence points towards a role of TBS in fracture risk prediction in patients with T2DM [21]. However, there are limited prospective data on TBS modifications in patients with T2DM when treated with different therapies for osteoporosis or diabetes, and to what extent TBS modifications influence fracture risk [22].

Clinical determinants of poor bone strength in diabetes are not fully understood. For instance, the association between glycaemic control and bone fragility is controversial. Poor glycaemic control has been associated with lower bone mineralization in most studies [23,24], but not in others [25]. It is known that some antidiabetic drugs affect bone metabolism. Sulfonylureas, insulin treatment and thiazolidinediones determine an increase in fracture risk [26,27,28]. In contrast, the effect of other antidiabetic drugs on fracture risk remains unclear [29].

The aim of our study was to investigate bone mineralization and micro-architecture in women with T2DM and good glycemic control who are treated with metformin in monotherapy and to explore non-glycemic determinants of bone alterations.

## 2. Results

### 2.1. Characteristics of the Study Population

This study involved 126 female patients with T2DM referred to our Endocrinology and Diabetes outpatient clinic of Sapienza University, Rome, Italy, for metabolic evaluations (mean age 62.96 ± 6.73 years, BMI 31.63 ± 7.09 kg/m^2^) who met the study’s inclusion criteria (Figure 1).

We also recruited 117 women without T2DM, comparable for age and body mass index (BMI) (mean age 61.91 ± 5.8 years, BMI 32.64 ± 7.12 kg/m^2^), as a comparison group. The main clinical features of the study population are shown in Table 1.

Women with T2DM compared with controls had significantly higher glycaemia (*p* < 0.001), HbA1c (*p* < 0.001), triglycerides (*p* = 0.001), gamma-glutamyl transferase (GGT) (*p* = 0.005) and parathyroid hormone (PTH) (*p* = 0.018). However, they showed lower waist circumference (*p* = 0.015) and lower levels of total, high-density lipoprotein (HDL-c) and low-density lipoprotein (LDL-c) cholesterol (*p* < 0.001). Statin treatment was significantly more frequent in patients with T2DM (*p* = 0.002). Similar prevalence of vitamin D supplementation (consisting of oral colecalcipherol, Vitamin D3) treatment was reported in women with and without T2DM.

### 2.2. DXA Results

The T2DM patients showed higher BMD and T-scores at all levels, and lower TBS than non-diabetic controls, as shown in Table 2. When dividing the study population according to TBS cut-offs [30], we observed that the prevalence of normal bone texture was significantly lower in the T2DM group compared with controls (*p* = 0.01).

The presence of T2DM was associated with TBS in the lowest tertile vs. highest tertile with adjusted OR: 2.47 (C.I. 95%: 1.19–5.16, *p* = 0.016). Conversely, diabetes status was confirmed to correlate with greater lumbar spine BMD (OR: 0.43, C.I. 95%: 0.21–0.89, *p* = 0.024), but not with BMD values measured in the other districts (femoral neck OR: 0.65, C.I. 95%: 0.22–1.92, *p* = 0.433; total hip BMD OR: 0.56, C.I. 95%: 0.28–1.12).

### 2.3. Metabolic Correlates of Impaired TBS in T2DM Patients

In the univariate regression analyses, in T2DM women, TBS positively correlated with HDL-c (Standardized β coefficient: 0.21, *p* = 0.029) and Vitamin D (Standardized β coefficient: 0.25, *p* = 0.017) and negatively with the presence of menopausal status (Standardized β coefficient: −0.35, *p* < 0.001), number of components of metabolic syndrome (Standardized β coefficient: −0.23, *p* = 0.014), insulin resistance (HOMA-IR, Standardized β coefficient: −0.19, *p* = 0.014), BMI (Standardized β coefficient: −0.25, *p* = 0.005), and waist circumference (Standardized β coefficient: −0.33, *p* < 0.001). Similar results were obtained when performing the analyses in the whole study population (Appendix A). Lower HDL-c levels represented the major determinant of reduced TBS, independently of possible confounders, such as age, menopausal status, BMI, waist circumference, statin treatment, vitamin D, and physical activity, in the multivariate linear regression model (Table 3).

## 3. Discussion

In this study, we evaluated bone health in a population of T2DM women compared with women without diabetes comparable for age and BMI, recruited as a control group. We then explored metabolic predictors of bone quality in T2DM. We demonstrated that women with T2DM have more degraded bone texture, as assessed by TBS, compared with healthy women, also in the presence of similar or even increased BMD.

We showed that in women with T2DM, HDL cholesterol is the main predictor of bone alterations, independently of possible confounders such as age, BMI, waist circumference, menopausal status, physical activity, statin treatment and vitamin D levels. Previous reports investigating the association between lipid profile and bone metabolism in terms of BMD and fracture risk have shown inconsistent results, likely due to differences in study design, populations and methodology applied [31]. There are a few studies that have assessed the possible association between lipid profile and bone quality as evaluated by TBS, but not conducted in patients with T2DM. Panahi et al. showed a negative association between TBS and HDL-c in elderly men, but not in women [32]. Recently, an observational study conducted in postmenopausal women demonstrated that the triglycerides/HDL-c ratio was independently associated with degraded bone texture [33]. To our knowledge, there are no studies that have investigated the association between qualitative bone alterations and the metabolic profile in T2DM.

Several pathophysiological mechanisms may explain the association between low HDL-c levels and impaired trabecular bone structure, as shown in Figure 2. Firstly, adipose and bone tissues share a common progenitor: the multipotent mesenchymal stem cells (MSCs) in the bone marrow (BM), which can differentiate into osteoblasts, adipocytes and chondrocytes, depending on the characteristics of the medullary microenvironment [34]. Chronic immuno-inflammatory changes in BM influence the differentiation of MSCs into adipocytes and inhibit osteogenesis [35]. Moreover, the activation of peroxisome proliferator-activated receptor gamma (PPAR-γ) by lipid metabolites could also play a pivotal role in changes in bone in patients with insulin resistance and T2DM. PPAR-γ, which is activated by lipid oxidation products, inhibits osteoblast’s differentiation and promotes adipogenesis, causing bone changes in patients with metabolic disorders [31].

Animal models have shown interesting data linking bone and lipid metabolism. Martineau et al. investigated the role of Scavenger receptor class B type I (Scarb1), the principal HDL-c receptor, in bone metabolism. Scarb1-deficient mice showed increased HDL-c levels, higher bone volume and number of trabeculae, and enhanced bone formation [36]. In contrast, mice lacking Apolipoprotein A1 (Apo A1), which is the main protein component of HDL-c particles [37], displayed reduced bone formation and mass, lower osteoblastogenesis, higher adipogenesis and increased BM adiposity compared with the wild-types [38]. The mechanism proposed is the increased committed adipoblasts in the overall mesenchymal stem cell (MSC) pool, as shown by the greatly increased PPAR-γ mRNA levels. Thus, HDL-c, probably through Apo A1, have a central role in the regulation of bone remodeling and maintenance of bone quality. Apo A1 deficiency and consequently low HDL-c levels reduce the capacity of MSC to differentiate toward osteoblasts but promote adipogenesis, affecting bone quality and stability.

Finally, growing evidence is suggesting that inflammation also plays a fundamental role in the development of osteoporosis. Data indicate that pro-inflammatory cytokines, such as IFN-γ, IL-17A, IL-15 and TNFα, promote osteoclastogenesis and impair osteoblastogenesis [39]. Interestingly, HDL-c is able to inhibit the interaction between T lymphocytes and antigen presenting cells (APCs), preventing the activation of the latter and associated TNFα and IL-1β production [40] and suppressing gene expression of mediators of the type I interferon response pathway [41]. Therefore, HDL-c may indirectly influence bone metabolism due to its systemic and tissue-specific anti-inflammatory properties [42]. Figure 2 summarizes the main mechanisms potentially linking low HDL and impaired bone metabolism.

The association between glycaemic control and bone fragility is controversial [23,24,25]. It is well known that some antidiabetic drugs affect bone metabolism, increasing fracture risk (sulfonylureas, thiazolidinediones and insulin treatment) [26,27,28]. On the other hand, the effect of Glucagon-Like Peptide 1 Agonists (GLP1a) and Dipeptidyl Peptidase 4 Inhibitors (DPP4i) on fracture risk remains unclear, whereas metformin and Sodium-Glucose Cotransporter 2 Inhibitors (SGLT2i) seem to have a neutral action [43,44,45,46]. In order to identify non-glycaemic predictors of altered bone metabolism, our study was designed to minimize the influence of glycaemic control and different anti-diabetic therapies on bone health by selecting patients in good glycemic control (HbA1c ≤ 53 mmol/mol) and who were treated only with metformin without a history of other antidiabetic treatments.

Besides being affected by glycaemic control and antidiabetic agents, bone health can also be influenced by lifestyle interventions [47]. Diet and physical activity can ameliorate glycaemic control and favour weight loss [48]. Similarly exercise and consumption of dairy products have beneficial effects on bone health [49,50]. To assess the effect of physical activity in both glycaemic, lipid and bone metabolism, we administered the International Physical Activity Questionnaire [51] to all study participants and included physical activity intensity in the stepwise multivariate linear regression model.

Bone fragility in diabetes results, not only from alterations in bone mineralization, but also from alterations in bone microstructure [19] that can be indirectly assessed by TBS. Nearly all the subjects included in our study were overweight or obese, so higher BMD and lower TBS are influenced by higher BMI [52,53], which leads to errors in DXA measurements caused by soft tissue thickness [54]. Therefore, we included controls with a comparable BMI distribution to our study’s population. Nevertheless, T2DM women showed higher BMD in all scans compared with controls. Although the mean TBS value was similar in T2DM patients and controls, when dividing the two cohorts into subgroups according to TBS cut-offs, we observed that the prevalence of individuals in the T2DM group with normal bone structure was less than half of that in the controls. This was also in the presence of normal or even increased BMD, in line with other studies [21,55]. Similarly, we found that the presence of T2DM was associated with TBS values in the lowest tertile, compared with the highest tertile, with OR 2.47 (C.I. 95%: 1.19–5.16) adjusted for age, menopausal status, and BMI.

This study has some limitations. First, the cross-sectional design does not allow us to establish a causal nexus mediating the association between low HDL-c and impaired TBS in T2DM. However, the experimental evidence on the HDL-c anti-inflammatory properties and its beneficial interaction with bone precursor cells may point to a causal role of reduced HDL-c in increased bone fragility in metabolic diseases. Studies with longitudinal designs on larger populations are warranted to further explore this novel finding. Secondly, TBS does not represent the gold standard technique for evaluating bone microarchitecture as it is an indirect estimator of lumbar bone structure based on analyzing pixel gray-level variations. Data obtained with the HRpQCT technique, which explores 3D cortical and trabecular bone quality at the radius and tibia levels, would provide the best estimation of bone architecture; however, this is a costly method not readily available in clinical practice.

All the T2DM patients were on metformin treatment, as specified by the inclusion criteria, and one in four had been taking vitamin D supplementation at the time of the study recruitment. Although we cannot exclude an influence of these agents on bone outcomes, as metformin use at a stable dose in the previous 3 months was reported in the entire T2DM cohort, it is unlikely that this agent had an influence on the association between low HDL-c and impaired TBS in these patients. As for vitamin D, the percentage of study participants among the T2DM women and control group taking vitamin D supplementation at the time of study enrolment was comparable; moreover, the study results were adjusted for serum vitamin D levels, which did not impact the association between impaired TBS and low HDL-c in the multivariate regression analysis.

Finally, TBS may represent a valuable tool in the prediction of bone fragility in diabetes, in addition to BMD. In fact, some studies have shown that TBS is able to predict osteoporotic fractures in patients with T2DM [21,56]. Further studies are needed to validate TBS’s use in clinical practice to better assess fracture risk in patients with T2DM.

## 4. Materials and Methods

### 4.1. Study Population

For this observational case-control study, we recruited 126 eligible women with T2DM consecutively referred to our Endocrinology and Diabetes outpatient clinic at Sapienza University, Rome, Italy, for metabolic evaluations (mean age 62.96 ± 6.73 years), who met the following inclusion criteria: age 18 or older; T2DM diagnosis according to American Diabetes Association (ADA) Guidelines [57]; BMI between 20 and 40 kg/m^2^ and body weight ≤ 120 kg (due to limitations of the DXA equipment); HbA1c < 53 mmol/mol; and metformin monotherapy at a stable dose for at least 3 months. As a reference group, we also recruited 117 healthy women with age and BMI distribution comparable to that of the T2DM cohort (mean age: 61.91± 5.8 years, BMI: 32.64 ± 7.12 kg/m^2^).

The main exclusion criteria were as follows: drugs affecting bone metabolism (bisphosphonates, hormone replacement therapy, calcitonin, corticosteroids etc.); secondary causes of osteoporosis (Paget’s disease, osteomalacia, hyperparathyroidism, hyperthyroidism, liver or kidney failure) and current or history of therapy with antidiabetic agents other than metformin.

### 4.2. Metabolic Evaluations

Each participant underwent a medical history collection and physical examination at the study site. All ongoing therapies were recorded, including Vitamin D and calcium supplements. Weight (kg), height (m) and waist circumference (cm) were measured. Blood pressure (mmHg) was assessed after 5 min of resting, using the average of the second and third measurement in the analysis. To assess physical activity intensity, the International Physical Activity Questionnaire [51] was administered to all participants.

Fasting blood samples were drawn and the following blood tests were performed by centralized standard methods at Sapienza University: FBG (mg/dL), insulin (IU/mL), HbA1c (mmol/mol), total cholesterol (mg/dL), HDL-C cholesterol (mg/dl), triglycerides (mg/dL), ALT(IU/L), AST(IU/L), GGT (mg/dL), serum creatinine (mg/dL), TSH (mU/L), PTH (pg/mL), vitamin D (ng/mL), calcium (mg/dL) and phosphate (mg/dL). LDL levels were obtained using the Friedewald formula; insulin resistance was estimated using HOMA-IR.

### 4.3. DXA and TBS Assessment

We assessed BMD at the lumbar spine (L1–L4 anteroposterior) and hip (total hip and femoral neck). Scans were performed by densitometers Hologic Discovery A (S/N 84191, Bedford, MA, USA) in the Bone Metabolism Service of Sapienza University of Rome by a trained technician using standardized procedures. BMD was expressed in g/cm^2^. According to the WHO criteria, osteoporosis was diagnosed in subjects with a T-score ≤ −2.5, and osteopenia in those with a T-score between −2.5 and −1.0 [58].

The integrated software TBS iNsight, version 2.1.2.0, was applied by the same technician to the site-matched spine scans for the evaluation of TBS. We used the following cut-off points for TBS evaluation as described by McCloskey et al. [30]: TBS > 1.31 as normal, TBS between 1.23 and 1.31 for partially degraded bone texture, and TBS < 1.23 indicating degraded texture.

### 4.4. Statistics

Descriptive statistics are presented as mean ± standard deviation (SD) for continuous variables or percentage for categorical variables. Differences between independent groups were compared by Student’s *t*-test for continuous variables and by χ^2^ test for categorical variables.

To test the association between presence of diabetes and bone health, we stratified the whole study population in tertiles according to TBS and BMD (total hip, femoral neck and lumbar spine) and calculated odds ratios for the lowest vs. highest tertile with 95% confidence intervals (CIs) associated with diabetes status, by logistic regression analyses adjusted for age, menopausal status, and BMI. Moreover, the prevalence of individuals with normal vs. degraded bone structure within the T2DM group and controls was calculated by a χ^2^ test.

In order to identify predictors of impaired TBS in T2DM, TBS was analysed as the dependent variable in univariate analyses, and a multivariate model was built that included variables significantly associated with TBS in the univariate analyses and potential clinical confounders. R^2^ was calculated as a goodness-of-fit measure, and both R and R^2^ were reported. A *p* value < 0.05 was taken to indicate a statistically significant effect. Statistical analyses were performed using IBM Corp. Released 2020. IBM SPSS Statistics for Macintosh, Version 27.0. Armonk, NY: IBM Corp [59].

Sample size and power calculation. The sample size of this investigation was calculated according to data from the cross-sectional study by Leslie WD and coinvestigators [21]. In a BMI-, age- and disease-adjusted analysis, they found that mean ± SD TBS = 1.245 ± 0.125 in non-diabetic women vs. TBS = 1.194 ± 0.112 in T2DM women (see also [60]). Based on these findings, *n* = 116 individuals per group were needed for sufficient statistical power to detect the effect of T2DM on TBS, with an α-error = 0.05 and power = 90%.

## 5. Conclusions

Our study demonstrates that women with T2DM have altered bone microstructure as evaluated by TBS, even though they show normal or increased bone mineral mass. The specific increase in BMD impairs the classical diagnosis of osteoporosis in the absence of fragility fractures. Therefore, TBS is a useful tool in addition to BMD for assessing bone health in patients with T2DM in clinical practice. To our knowledge, our study is the first to have investigated the association between qualitative bone alterations and lipid metabolism in T2DM and to have demonstrated that low HDL-c is significantly and independently associated with degraded bone quality in this population. The measurement of HDL-c is highly utilized by clinicians to help predict cardiovascular risk, especially in patients with T2DM. Therefore, low HDL-c could be considered not only to detect patients at high cardiovascular risk, but also to alert clinicians to the presence of an impaired trabecular bone structure in women and increased fracture risk. Further studies are needed to confirm these results and continue to deepen the knowledge of the underlying links between lipid and bone metabolism.

## Figures and Tables

**Figure 1 ijms-24-06474-f001:**
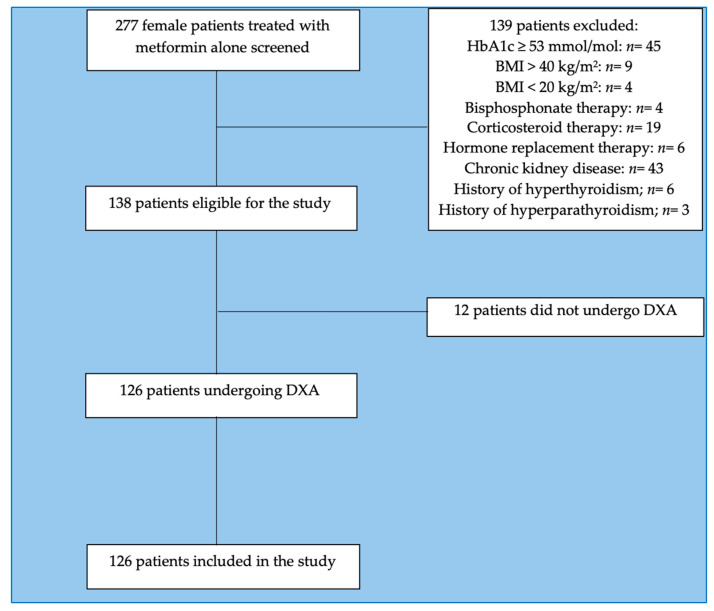
CONSORT flow-chart of the study. Abbreviations: BMI: body mass index, HbA1c: glycosylated hemoglobin.

**Figure 2 ijms-24-06474-f002:**
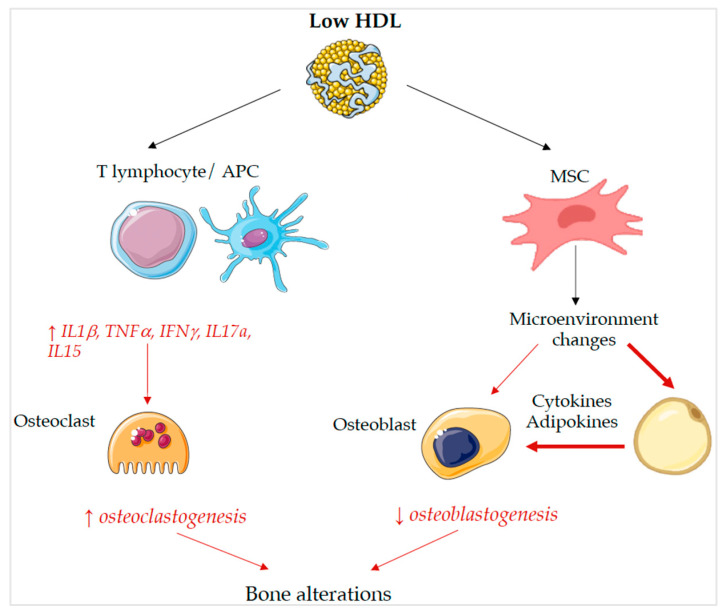
Molecular mechanisms that link low HDL-c levels to bone alterations. For full description, see text. Abbreviations: HDL: high-density lipoprotein cholesterol; APC: antigen presenting cell; MSC: multipotent mesenchymal stem cell.

**Table 1 ijms-24-06474-t001:** Differences in clinical and biochemical features between T2DM patients and controls.

Clinical Parameters	T2DM (*n* = 126)	Controls (*n* = 117)	*p* Value
Age (years)	63 ± 6.7	61.9 ± 5.8	0.20
BMI (kg/m^2^)	31.6 ± 7.1	32.6 ± 7.1	0.27
Waist circumference (cm)	106.2 ± 15.7	111.6 ± 14.6	0.015
Menopause onset age (years)	49.2 ± 4.6	49.1 ± 4.7	0.81
SBP (mmHg)	131.4 ± 20.8	133.1 ± 15.4	0.49
DBP (mmHg)	82.5 ± 9.1	81.9 ± 11.3	0.66
FBG (mg/dL)	117.5± 19.6	88.3 ± 8.2	<0.001
HbA1c (mmol/mol)	45.5 ± 0.1	35.3 ± 0.1	<0.001
Total cholesterol (mg/dL)	182.8 ± 35.5	209.7 ± 35.8	<0.001
HDL-C-c (mg/dL)	53.4 ± 13.3	59.7 ± 14.4	<0.001
LDL-c (mg/dL)	104.6 ± 33.8	128.7 ± 32.8	<0.001
Triglycerides (mg/dL)	135.4 ± 62.3	111.7 ± 47.3	0.001
AST (IU/L)	22.8 ± 13.5	20.6 ± 9.4	0.18
ALT (IU/L)	25.1 ± 15.6	22.4 ± 11.7	0.16
GGT (IU/L)	31.9 ± 30	20.7 ± 18	0.005
Serum Creatinine (mg/dL)	0.7 ± 0.2	0.7 ± 0.1	0.83
TSH (mU/L)	2.1 ± 1.3	2 ± 2	0.87
PTH (pg/mL)	57 ± 39.8	46 ± 22	0.018
Vitamin D (ng/mL)	24.7 ± 16.1	23.4 ± 13.6	0.54
Calcium (mg/dL)	9.5 ± 0.4	9.4 ± 0.5	0.13
Phosphate (mg/mL)	3.6 ± 0.5	3.6 ± 0.5	0.47
Presence of menopausal state	92.1%	97.4%	0.063
Physical activity			
None-low intensity	46%	46.1%	0.53
Moderate intensity	46%	43.6%	
High intensity	8%	10.3%	
Presence of statin treatment	42.2%	21.3%	0.002
Atorvastatin	37.7%	40%	
Simvastatin	34%	28%	0.56
Rosuvastatin	28.3%	32%	
Presence of Vitamin D supplementation	25%	22.2%	0.64

Data are shown as mean values ± standard deviation (SD) or percentages. Significant *p* values are reported in bold. Abbreviations: T2DM: type 2 diabetes mellitus; BMI: body mass index; SBP: systolic blood pressure; DBP: diastolic blood pressure; FBG: fasting blood glucose; HbA1c: glycosylated hemoglobin; HDL-C: high-density lipoprotein; LDL: low-density lipoprotein; AST: aspartate aminotransferase; ALT: alanine aminotransferase; GGT: gamma-glutamyl transferase; TSH: thyroid stimulating hormone; PTH: parathyroid hormone.

**Table 2 ijms-24-06474-t002:** Differences in bone parameters between patients with T2DM and controls.

DXA-Derived Parameters	T2DM (*n* = 126)	Controls (*n* = 117)	*p* Value
Lumbar spine BMD (g/cm^2^)	1.010 ± 0.165	0.937 ± 0.161	0.031
Lumbar spine T-score	−0.620 ± 1.258	−0.787 ± 1.524	0.359
Total hip BMD (g/cm^2^)	0.936 ± 0.131	0.891 ± 0.145	0.080
Total hip T-score	−0.384 ± 0.961	−0.491 ± 1.132	0.801
Femoral neck BMD (g/cm^2^)	0.744 ± 0.125	0.725 ± 0.119	0.391
Femoral neck T-score	−1.002 ± 1.137	−1.213 ± 0.903	0.373
TBS	1.180 ± 0.112	1.209 ± 0.120	0.060
Individuals with degraded bone architecture (TBS ≤ 1.23)	62.7%	59.5%	0.61
Individuals with partially degraded bone architecture (1.23 < TBS ≤ 1.31)	30.2%	22.4%	0.17
Individuals with normal bone architecture (TBS > 1.31)	7.1%	18.1%	0.01

Data are shown as mean values ± standard deviation (SD) or percentages. Abbreviations: BMD: bone mineral density; DXA: dual-energy X-ray absorptiometry; TBS: trabecular bone score; T2DM: type 2 diabetes mellitus.

**Table 3 ijms-24-06474-t003:** Multivariate linear regression analyses for TBS value.

Variables	Unstandardized Coefficient		Standardized Coefficient	*p* Value
	β	Standard Error	β	
Age	−0.004	0.005	−0.162	0.464
Menopausal state	0.008	0.005	0.303	0.175
BMI	−0.007	0.008	−0.304	0.411
Waist circumference	0.002	0.004	0.199	0.634
HDL-c	0.006	0.002	0.603	0.028
Physical activity	−0.056	0.069	−0.214	0.432
Statin treatment	0.016	0.061	0.066	0.793
Vitamin D	0.002	0.061	0.066	0.793
Constant	0.686	0.514	-	0.205

TBS is the dependent variable. Model’s R: 0.682, R^2^: 0.465. Abbreviations. BMI—body mass index; HDL-c—high-density lipoprotein cholesterol.

## Data Availability

The data presented in this study are available on request from the corresponding author. The data are not publicly available due to privacy restrictions and lack of specific patient consent.

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
