# Peer review of "Reduced High-Density Lipoprotein Cholesterol Is an Independent Determinant of Altered Bone Quality in Women with Type 2 Diabetes"

_ijms, 2023, doi:10.3390/ijms24076474_

Round 1

Reviewer 1 Report (New Reviewer)

 This relatively small cross-sectional study makes an interesting observation regarding the presence of a positive correlation between HDL cholesterol and lumbar spine trabecular bone score (TBS) in women with a type 2 diabetes. The authors speculate on the pathophysiologic explanation for this and the existence of common pathways between metabolic syndrome and bone health in diabetes.

Major comments:

1.       I have major concerns regarding the confounding effect of abdominal tissue thickness on TBS and HDL cholesterol. TBS measurements appear lower in individuals with greater abdominal tissue thickness (strongly related to waist circumference) which is only partially accounted for through BMI adjustment. HDL cholesterol is also negatively associated with weight circumference as shown in large epidemiologic studies (1). The authors report that the bivariate negative association between TBS and waist circumference (standardized beta -0.33) is stronger than the positive association with HDL-C (standardized beta 0.21). It is therefore essential that any associations between TBS and HDL cholesterol be adjusted for confounding waist circumference. If the results in Table 3 significantly change after adjustment for waist circumference then it would likely require extensive changes to the manuscript, discussion and conclusions.

2.       The title, discussion and conclusion refer to "bone microarchitecture" and "bone microstructure." The ability of TBS to assess these parameters is controversial and TBS is best considered as a BMD-independent texture measurement that correlates with fracture risk.  Moreover, available evidence suggest that trabecular micro architecture measured from HRpQCT is not adversely affected by type 2 diabetes (2-5).

Minor comments:

1.       The introduction states that more than 1 out of 3 postmenopausal positive women has osteoporosis. This is unusually high – the study cited is from Korea and is not applicable to other populations.

2.       The introduction has typographical errors, "loose more bone mineral density" versus "lose more bone mineral density"; "referring to our endocrinology and diabetes outpatient clinic" versus "referred to".

3.       The introduction cites reference 10 for accelerated BMD loss at the femoral neck in type 2 diabetes, which was quite marginal, fails to note the higher initial BMD such that individuals with diabetes remain at low fracture for osteoporosis. The article cited actually concludes: “our findings suggest that diabetes is associated with a slightly greater rate of BMD loss at the femoral neck but this small difference is unlikely to be clinically significant or explain the increased fracture risk that has been associated with type 2 diabetes. No significant effect of diabetes was seen on BMD loss at the total hip or lumbar spine.”

4.       The estimate of 70% increased risk for fragility fracture in type 2 diabetes is higher than most meta- analyses. A more recent meta-analysis found “T2DM induced excess risk to all fractures (OR: 1.19; 95% CI 1.09–1.31; P < 0.001)” (6)

5.       Measurements in Table 1 are provided to 2-decimal accuracy which is excessive for many variables such as age, BMI, waist circumference, etc.

6.       The authors refer to "validated TBS cutoffs" citing an outdated 2014 reference which used cutoffs were not validated in any objective sense.  Most now used the tertile cutoffs from McCloskey at al (7).

7.       The caption to Figure 1 refers to "altered bone structure" which is a vague term. I think this refers to table 2 "individuals with normal bone architecture". Regardless, the figure duplicates data available in the table and should be deleted.

8.       Section 2.3 reports the BMI standardized beta coefficient is 0.25; should this be -0.25? This section needs to report associations with weight circumference.

9.       The discussion notes "higher BMD is influenced by higher BMI." BMI also affects TBS.

10.   One of the metabolic evaluations listed in section 4.2 is HOMA-IR but does not appear to be reported or used in any analysis.

11.   Section 4.3 describes the densitometers used as both GE and HologicA/QDR 4500W. This is incorrect; GE and Hologic are the major manufactures; QDR-4500 is a Hologic product. Please clarify.

12.   The controls have greater waist circumference than women with T2D which is counterintuitive. Is there a mistake?

References

1.            Nevill AM, Duncan MJ, Myers T. BMI is dead; long live waist-circumference indices: But which index should we choose to predict cardio-metabolic risk? Nutr Metab Cardiovasc Dis. Jul 2022;32(7):1642-50. Epub 2022/05/08.

2.            Nilsson AG, Sundh D, Johansson L, Nilsson M, Mellstrom D, Rudang R, et al. Type 2 Diabetes Mellitus Is Associated With Better Bone Microarchitecture But Lower Bone Material Strength and Poorer Physical Function in Elderly Women: A Population-Based Study. J Bone Miner Res. May 2017;32(5):1062-71.

3.            Samelson EJ, Demissie S, Cupples LA, Zhang X, Xu H, Liu CT, et al. Diabetes and Deficits in Cortical Bone Density, Microarchitecture, and Bone Size: Framingham HR-pQCT Study. J Bone Miner Res. Jan 2018;33(1):54-62.

4.            Starr JF, Bandeira LC, Agarwal S, Shah AM, Nishiyama KK, Hu Y, et al. Robust Trabecular Microstructure in Type 2 Diabetes Revealed by Individual Trabecula Segmentation Analysis of HR-pQCT Images. J Bone Miner Res. Sep 2018;33(9):1665-75. Epub 2018/05/12.

5.            Walle M, Whittier DE, Frost M, Muller R, Collins CJ. Meta-analysis of Diabetes Mellitus-Associated Differences in Bone Structure Assessed by High-Resolution Peripheral Quantitative Computed Tomography. Curr Osteoporos Rep. Oct 3 2022. Epub 2022/10/04.

6.            Dou J, Wang J, Zhang Q. Differences in the roles of types 1 and 2 diabetes in the susceptibility to the risk of fracture: a systematic review and meta-analysis. Diabetol Metab Syndr. Aug 16 2021;13(1):84. Epub 2021/08/18.

7.            McCloskey EV, Oden A, Harvey NC, Leslie WD, Hans D, Johansson H, et al. A Meta-Analysis of Trabecular Bone Score in Fracture Risk Prediction and Its Relationship to FRAX. J Bone Miner Res. May 2016;31(5):940-8.

Author Response

Reviewer 1 - Comments and Suggestions for Authors

 This relatively small cross-sectional study makes an interesting observation regarding the presence of a positive correlation between HDL cholesterol and lumbar spine trabecular bone score (TBS) in women with a type 2 diabetes. The authors speculate on the pathophysiologic explanation for this and the existence of common pathways between metabolic syndrome and bone health in diabetes.

We thank the reviewer for his/her encouraging comments which allowed to improve our manuscript and to reinforce the robustness of our study findings. Please find below the responses to all the points raised during the revision.

Major comments:

  1. I have major concerns regarding the confounding effect of abdominal tissue thickness on TBS and HDL cholesterol. TBS measurements appear lower in individuals with greater abdominal tissue thickness (strongly related to waist circumference) which is only partially accounted for through BMI adjustment. HDL cholesterol is also negatively associated with weight circumference as shown in large epidemiologic studies (1). The authors report that the bivariate negative association between TBS and waist circumference (standardized beta -0.33) is stronger than the positive association with HDL-C (standardized beta 0.21). It is therefore essential that any associations between TBS and HDL cholesterol be adjusted for confounding waist circumference. If the results in Table 3 significantly change after adjustment for waist circumference then it would likely require extensive changes to the manuscript, discussion and conclusions.

As pointed out by the reviewer, in the univariate regression analyses, lower TBS was associated with more elevated BMI and waist circumference. However, initially we did not include these parameters together in the same multivariate model for the risk of collinearity, since BMI and waist circumference usually are strongly dependent with each other (b coefficient= 0.89, p<0.001). Nonetheless, we do see this reviewer’s point and have now entered also waist circumference in a stepwise multivariate model, which identifies the best predictor from a larger set of potential covariates. This new analysis reinforces the finding of an association between lower HDL and impaired TBS after adjusting for multiple confounders, such as age, menopausal state, HDL, physical activity, BMI, waist circumference, vitamin D and statin treatment (standardized b: 0.44, p<0.009, model’s R: 0.635, R2: 0.403). We thank the reviewer for this important comment and have now substituted the old model with the one suggested (new Table 3). Results’ description has been modified accordingly: “Lower HDL-c levels represented the major determinant of reduced TBS, independently of possible confounders, such as age, menopausal status, BMI, waist circumference, statin treatment, vitamin D, and physical activity, in the stepwise multivariate linear regression model (Table 3)” (Page 5, lines 182-186).

Table 3. Multivariate linear regression analyses for TBS value.

Variables

Non standardized

b coefficient

Standardized

b coefficients

P value

Constant

0.921

-

<0.001

HDL-c

0.005

0.49

0.009

Variables excluded

Variables

b in

P value

Physical activity

0.118

0.558

Age

-0.414

0.027

BMI

-0.143

0.477

Menopausal status

-0.464

0.012

Waist circumference

-0.132

0.521

Statin treatment

-0.187

0.350

Vitamin D

0.188

0.341

TBS is the dependent variable. model’s R: 0.635, R2: 0.403.

Abbreviations. BMI- body mass index; HDL-c- high-density lipoprotein cholesterol.

  1. The title, discussion and conclusion refer to "bone microarchitecture" and "bone microstructure." The ability of TBS to assess these parameters is controversial and TBS is best considered as a BMD-independent texture measurement that correlates with fracture risk.  Moreover, available evidence suggest that trabecular micro architecture measured from HRpQCT is not adversely affected by type 2 diabetes (2-5).

We thank the reviewer for this comment and have now addressed the controversy of trabecular bone alterations in conditions of type 2 diabetes in the study’s introduction adding the following sentences: “Data indicate the presence of cortical bone alterations in patients with T2DM [17], but the effects on trabecular bone remain controversial. In fact, studies conducted on bone biopsies of T2DM have shown the presence of fewer and thinner trabeculae leading to increased bone fragility, both in rats [18] and humans [19]. Conversely, bone trabecular microarchitecture was shown to be preserved or even improved in diabetic patients when explored with other imaging techniques, such as the High Resolution Peripheral Quantitative Computed Tomography (HR-pQCT) [17, 20]” (Page 2, lines 64-70).

As pointed out by the reviewer, the TBS does not represent the gold standard technique for evaluating bone microarchitecture, although it can provide an indirect estimation of 3D trabecular bone structure starting from 2D lumbar spine DXA images, analyzing pixel gray-level variations. HRpQCT instead assesses bone microarchitecture using a 3D imaging technique and allows to evaluate several parameters of cortical and trabecular bone quality at the radius and tibia level. Nevertheless, TBS has been associated not only with fracture risk, but also with trabecular indices derived from HRpQCT (1, 2) and bone biopsies (3, 4) in many populations. Indeed, these two methods offer different and complementary information on bone quality.

We thank the reviewer for this important input and have now added the following sentence to emphasize this study limitation in the new version of this manuscript. “Secondly, TBS does not represent the gold standard technique for evaluating bone microarchitecture, as it is an indirect estimator of lumbar bone structure analyzing pixel gray-level variations. Data obtained with the HRpQCT technique, which explores 3D cortical and trabecular bone quality at the radius and tibia levels, would provide the best estimation of bone architecture; however, this is a costly method not readily available in clinical practice” (Page 8, lines 319-325).

Minor comments:

  1. The introduction states that more than 1 out of 3 postmenopausal positive women has osteoporosis. This is unusually high – the study cited is from Korea and is not applicable to other populations.

This information has been substituted with the newest data reported by the observatory of the Italian Health Ministry in 2023: “In Italy, only 8.1% of the population report having osteoporosis, and the prevalence progressively increases with advancing age, particularly in females, affecting 47% of females aged > 74 years [4]” (Page 1, line 45, page 2 lines 46-47).

  1. The introduction has typographical errors, "loose more bone mineral density" versus "lose more bone mineral density"; "referring to our endocrinology and diabetes outpatient clinic" versus "referred to".

We apologise for the mistakes. These errors have been now amended (Page 9, line 343). We also checked again and corrected all the typos that we could find throughout the manuscript.

  1. The introduction cites reference 10 for accelerated BMD loss at the femoral neck in type 2 diabetes, which was quite marginal, fails to note the higher initial BMD such that individuals with diabetes remain at low fracture for osteoporosis. The article cited actually concludes: “our findings suggest that diabetes is associated with a slightly greater rate of BMD loss at the femoral neck but this small difference is unlikely to be clinically significant or explain the increased fracture risk that has been associated with type 2 diabetes. No significant effect of diabetes was seen on BMD loss at the total hip or lumbar spine.”

We thank the reviewer for this observation, and we have now removed this not fully appropriate reference from the reference list of our manuscript.

  1. The estimate of 70% increased risk for fragility fracture in type 2 diabetes is higher than most meta- analyses. A more recent meta-analysis found “T2DM induced excess risk to all fractures (OR: 1.19; 95% CI 1.09–1.31; P < 0.001)” (6)

We have now replaced this prevalence with data from the more recent meta-analysis indicated by the reviewer: “Patients with T2DM have 19% increased risk of any fragility fracture [10]” (Page 2, line 56).

  1. Measurements in Table 1 are provided to 2-decimal accuracy which is excessive for many variables such as age, BMI, waist circumference, etc.

We have amended this Table according to the reviewer’s comment.

  1. The authors refer to "validated TBS cutoffs" citing an outdated 2014 reference which used cutoffs were not validated in any objective sense.  Most now used the tertile cutoffs from McCloskey at al (7).

We have now repeated the statistical analyses using the TBS cut-offs suggested by the reviewer, which slightly differ from those used in the previous version of this manuscript. Results from these new analyses reinforced our previous finding of higher prevalence of women with normal TBS in non-diabetic vs T2DM group (18.1% vs 7.1%, p= 0.01) and are now reported in the Table 2.The reference for TBS stratification has been changed accordingly citing the work by McCloskey at al. (new reference 30).

TBS

Individuals with degraded bone architecture

 (TBS ≤ 1.23)

 Individuals with partially degraded bone architecture

 (1.23<TBS ≤ 1.31)

 Individuals with normal bone architecture

 (TBS > 1.31)

1.180 ± 0.112

62.7%

30.2%

7.1%

1.209 ± 0.120

59.5%

22.4%

18.1%

0.060

0.61

0.17

0.01

  1. The caption to Figure 1 refers to "altered bone structure" which is a vague term. I think this refers to table 2 "individuals with normal bone architecture". Regardless, the figure duplicates data available in the table and should be deleted.

Figure 1 has been deleted from the new version of the manuscript.

  1. Section 2.3 reports the BMI standardized beta coefficient is 0.25; should this be -0.25? This section needs to report associations with weight circumference.

We thank the reviewer for noticing this refuse and have now corrected this coefficient as -0.25. This section reports correlations between TBS and metabolic parameters, including BMI and waist circumference, in women with type 2 diabetes. Waist circumference has been now entered as an additional covariate in the multivariate linear regression model testing determinants of TBS in this population (see also answer to point 1).

  1. The discussion notes "higher BMD is influenced by higher BMI." BMI also affects TBS.

This paragraph has been modified according with this reviewer’s suggestion: “Nearly all subjects included in our study are overweight or obese, so higher BMD and lower TBS are influenced by higher BMI [52, 53], that leads to errors in DXA measurements caused by soft tissue thickness [54]. Therefore, we included controls with comparable BMI distribution to our study’s population(Page 8, lines 303-307).

  1. One of the metabolic evaluations listed in section 4.2 is HOMA-IR but does not appear to be reported or used in any analysis.

We have now added HOMA-IR among the variables tested for association with TBS and reported the result in section 2.3 (HOMA-IR: Standardized b coefficient: -0.19, p= 0.014).

  1. Section 4.3 describes the densitometers used as both GE and HologicA/QDR 4500W. This is incorrect; GE and Hologic are the major manufactures; QDR-4500 is a Hologic product. Please clarify.

We apologize for the refuse, the densitometers used is Hologic Discovery A (S/N 84191)Bedford, MA, USA), and have not corrected this information in the text (Page 10, lines 372-373).

  1. The controls have greater waist circumference than women with T2D which is counterintuitive. Is there a mistake?

We agree that these data may be counterintuitive, but this is not a mistake. Individuals in the control group have been selected to be comparable for age and BMI to T2DM patients, so they are also obese, but with normal glucose regulation, as for inclusion criteria. The finding of higher waist circumference in non-diabetic subjects may be driven by slightly higher BMI, although this difference is not statistically significant (mean±SD BMI, controls: 32.6 ± 7.1 Kg/m2, vs T2DM: 31.6 ± 7.1 Kg/m2).

Reviewer 2 Report (New Reviewer)

As I see, this is a revised version of your manuscript, the 13 authors succeeding in a main manuscript of 9 pages (with many empty lines and one irrelevant figure). Also, it can be seen that very few corrections / completions have been added. Please see my suggestions and main concerns regarding this manuscript:

§  Remove the abbreviation from the title.

§  L82. Aim of the study is poor. As the topic is not at all a new one, in the Aim of the study, please mention what do you consider that your research brings new to the field? Or the special aspects? Why have you chosen this topic if the literature is plenty of papers in same/similar topic? What is the relevance of publishing it?

§  All the Tables. In the head of each table, it must be completed what represent the data in the 1st column.

§  For the values presented in the Figure 2 is not necessary an entire figure. A sentence would be enough. The figure is not relevant.

§  L138-142. Information is in duplicate with that presented in the Figure 2. An info must be presented in a single form (which is most relevant). Also, many duplicated info in paragraph L161-170 and the Table 2 – please proceed consequently.

§  The Discussion chapter must be improved. It would be relevant in describing the role of antidiabetic medication in the risk reduction of osteoporosis (such as metformin, SGLT2 inhibitors, insulin). I suggest checking and referring to https://doi.org/10.3390/ijms24044029Also discuss certain preventive strategies for reduction of osteoporosis (https://doi.org/10.3390/jcm7100297in patients with type 2 diabetes (such as increasing the physical activity, diet, life style, etc).

§  Additionally, please make a special summarizing figure where you present the pathophysiological association between HDL-cholesterol and osteoporosis.

§  L274-282.The Inclusion/exclusion criteria would be more relevant in a CONSORT type flow chart.

§  Subsection 4.4. The computer softs used for statistic must be mentioned and referenced. I suggest checking and referring to https://libguides.library.kent.edu/statconsulting/software-help and proceed consequently.

§  The second Subsection. 4.4. (wrongly numbered) referring to Ethics standards, must be removed as duplicates the info of L369-371.

Conclusions does not brings any interesting aspect as to justify this publication.

Author Response

Reviewer 2 - Comments and Suggestions for Authors

As I see, this is a revised version of your manuscript, the 13 authors succeeding in a main manuscript of 9 pages (with many empty lines and one irrelevant figure). Also, it can be seen that very few corrections / completions have been added. Please see my suggestions and main concerns regarding this manuscript:

  1. Remove the abbreviation from the title.

The abbreviation has been removed from the title.

  1. Aim of the study is poor. As the topic is not at all a new one, in the Aim of the study, please mention what do you consider that your research brings new to the field? Or the special aspects? Why have you chosen this topic if the literature is plenty of papers in same/similar topic? What is the relevance of publishing it?

We acknowledge that, in the present form, the aim of this study might make appear all investigations as  confirmatory research. However, there are major novelties in this study, including the selection of patients in good glycaemic control and who are very homogeneous for diabetes treatment, since all were using metformin at the time of the enrolment, which does not affect bone metabolism. Also the investigation of the role of  non-glycaemic variables on bone alterations is original. These inclusion criteria have been chosen to minimize the impact of treatments/ high glucose levels on the primary outcome. We thank the reviewer and have now re-written the study aim focusing on these aspects: “The aim of our study was to investigate bone mineralization and micro-architecture in women with T2DM in good glycemic control and treated with metformin in mono-therapy, and to explore non-glycemic determinants of bone alterations” (Page 2, lines 84-86).

  1. All the Tables. In the head of each table, it must be completed what represent the data in the 1st column.

All tables have been completed with the definition of data in the 1st column.

  1. For the values presented in the Figure 2 is not necessary an entire figure. A sentence would be enough. The figure is not relevant.

Figure 2 has been removed from the new version of this manuscript, the results previously shown are described in the text: “The presence of T2DM was associated with TBS in the lowest tertile vs highest tertile with adjusted OR: 2.47 (C.I. 95%: 1.19-5.16, p= 0.016). Conversely, diabetes status was con-firmed to correlate with greater lumbar spine BMD (OR: 0.43, C.I. 95%: 0.21-0.89, p= 0.024), but not with BMD values measured in the other districts (femoral neck OR: 0.65, C.I. 95%: 0.22-1.92, p= 0.433; total hip BMD OR: 0.56, C.I. 95%: 0.28-1.12)”(Page 5, lines 169-173).

  1. L138-142. Information is in duplicate with that presented in the Figure 2. An info must be presented in a single form (which is most relevant). Also, many duplicated info in paragraph L161-170 and the Table 2 – please proceed consequently.

We have now removed Figure 2, as indicated in response to point 4. Information at L161-170 are not shown in Table 2 or other tables (“At the univariate regression analyses, in T2DM women TBS positively correlated with HDL-c (Standardized b coefficient: 0.21, p= 0.029) and Vitamin D (Standardized b coefficient: 0.25, p= 0.017) and negatively with the presence of menopausal status (Standardized b coefficient: -0.35, p< 0.001), number of components of metabolic syndrome (Standardized b coefficient: -0.23, p= 0.014), BMI (Standardized b coefficient: 0.25, p= 0.005), and waist circumference (Standardized b coefficient: -0.33, p< 0.001). Similar results were obtained when performing the analyses in the whole study population (Supplementary Table 1S).

We suppose that a formatting change shifted the line numbering, so that the lines indicated by the reviewer are not those that he/she meant. Indeed, we identified some results at lines which were also included in Table 2 and those have been now just summarized in the text, leaving all details in the table: “T2DM patients showed higher BMD and T-score at all levels, and lower TBS than non-diabetic controls, as showed in Table 2”(Page 4, lines 154-156).

  1. The Discussion chapter must be improved. It would be relevant in describing the role of antidiabetic medication in the risk reduction of osteoporosis (such as metformin, SGLT2 inhibitors, insulin). I suggest checking and referring to https://doi.org/10.3390/ijms24044029. Also discuss certain preventive strategies for reduction of osteoporosis (https://doi.org/10.3390/jcm7100297) in patients with type 2 diabetes (such as increasing the physical activity, diet, life style, etc).

We thank the reviewer for this suggestion and have now extended the Discussion adding the following paragraphs and relative references:It is well known that some antidiabetic drugs affect bone metabolism, increasing fracture risk (sulfonylureas, thiazolidinediones and insulin treatment) [26-28]. On the other hand, the effect of Glucagon Like Peptide 1 Agonists (GLP1a) and Dipeptidyl Peptidase 4 Inhibitors (DPP4i) on fracture risk remains unclear, whereas metformin and Sodium-Glucose Cotransporter 2 Inhibitors (SGLT2i) seem to have neutral actions [43-46]” (Page 8, lines 285-289).

“Besides being affected by glycaemic control and antidiabetic agents, bone health can also be influenced by lifestyle interventions [47]. Diet and physical activity can ameliorate glycaemic control and favour weight loss [48].  Similarly exercise and consumption of dairy products have beneficial effects on bone health [49, 50]. To assess the effect of physical activity both in glycaemic, lipid and bone metabolism, we administered the International Physical Activity Questionnaire [51] to all study participants and included physical activity intensity in the stepwise multivariate linear regression model” (Page 8, lines 294-300).

  1. Additionally, please make a special summarizing figure where you present the pathophysiological association between HDL-cholesterol and osteoporosis.

We have now added to our manuscript Figure 2, which help the readers understand the possible mechanisms linking bone and lipid metabolism.

Figure 2. Molecular mechanisms that link low HDL-c levels to bone alterations. For full de-scription, see text.

Abbreviations: HDL: high-density lipoprotein cholesterol; APC: Antigen Presenting Cell; MSC: Multipotent mesenchymal stem cell.

  1. L274-282.The Inclusion/exclusion criteria would be more relevant in a CONSORT type flow chart.

A CONSORT type flow chart has been now provided in the text, as new Figure 1.

Figure 1. CONSORT flow-chart of the study.

Abbreviations: BMI: body mass index, HbA1c: glycosylated hemoglobin.

  1. Subsection 4.4. The computer softs used for statistic must be mentioned and referenced. I suggest checking and referring to https://libguides.library.kent.edu/statconsulting/software-help and proceed consequently.

The computer softs used for statistic has been now mentioned and referenced as indicated by this reviewer:“Statistical analyses were performed using IBM Corp. Released 2020. IBM SPSS Statistics for Macintosh, Version 27.0. Armonk, NY: IBM Corp [new reference 59] (Page 10, lines 399-400).

  1. The second Subsection. 4.4. (wrongly numbered) referring to Ethics standards, must be removed as duplicates the info of L369-371.

The subsection on ethics standards has been removed, as suggested by the reviewer.

  1. Conclusions does not brings any interesting aspect as to justify this publication.

We have now revised our conclusions focusing on the novel findings of our investigation (section 5): “Our study demonstrates that women with T2DM have altered bone microstructure evaluated by TBS, even though they show normal or increased bone mineral mass. The specific increase in BMD impairs the classical diagnosis of osteoporosis in absence of fragility fractures. So, TBS is a useful tool in assessing bone health in addition to BMD in the clinical practice in patients with T2DM. To our knowledge, our study is the first one to have investigated the association between qualitative bone alterations and lipid metabolism in T2DM and to have demonstrated that low HDL-c is significantly and independently associated with degraded bone microarchitecture in this population. The measurement of HDL-c is highly utilized by clinicians to help predict cardiovascular risk, especially in patients with T2DM. Therefore, low HDL-c could be considered not only to detect patients at high cardiovascular risk, but also to alert clinicians of the presence of an impaired trabecular bone structure in women and increased fracture risk” (Page 10, lines 208-219).

Round 2

Reviewer 1 Report (New Reviewer)

The authors have not fully addressed the following comments:

1.       I have major concerns regarding the confounding effect of abdominal tissue thickness on TBS and HDL cholesterol. TBS measurements appear lower in individuals with greater abdominal tissue thickness (strongly related to waist circumference) which is only partially accounted for through BMI adjustment. HDL cholesterol is also negatively associated with weight circumference as shown in large epidemiologic studies. The authors report that the bivariate negative association between TBS and waist circumference (standardized beta -0.33) is stronger than the positive association with HDL-C (standardized beta 0.21). It is therefore essential that any associations between TBS and HDL cholesterol be adjusted for confounding waist circumference. If the results in Table 3 significantly change after adjustment for waist circumference then it would likely require extensive changes to the manuscript, discussion and conclusions.

Additional comment: I remain concerned that confounders, especially waist circumference, may significantly affect the findings and conclusions. The authors have not adequately controlled for confounders in the multivariate linear regression presented in Table 3. Apparently they performed stepwise selection but all variables were excluded except for HDL-C. They need to present a model that retains all covariates to support the statement that the effect of HDL-C is independent. If the authors are concerned about collinearities (for example BMI and weight circumference), then residuals can be used, but this should not affect the HDL-C/TBS association.

2.       The title, discussion and conclusion refer to "bone microarchitecture" and "bone microstructure." The ability of TBS to assess these parameters is controversial and TBS is best considered as a BMD-independent texture measurement that correlates with fracture risk.  Moreover, available evidence suggest that trabecular micro architecture measured from HRpQCT is not adversely affected by type 2 diabetes.

Additional comment: The authors have not assessed trabecular “microarchitecture” directly and the title is misleading. TBS reflects “altered bone texture”. I think “microarchitecture” should be removed from the title, aims, results, discussion for the reasons stated below.

3.       Section 2.3 reports the BMI standardized beta coefficient is 0.25; should this be -0.25?

Additional comment: This standardized beta coefficient does not appear to have been corrected

Author Response

  1. Additional comment: I remain concerned that confounders, especially waist circumference, may significantly affect the findings and conclusions. The authors have not adequately controlled for confounders in the multivariate linear regression presented in Table 3. Apparently they performed stepwise selection but all variables were excluded except for HDL-C. They need to present a model that retains all covariates to support the statement that the effect of HDL-C is independent. If the authors are concerned about collinearities (for example BMI and weight circumference), then residuals can be used, but this should not affect the HDL-C/TBS association.

In the stepwise multivariate model, HDL-C was the only variable retained in the model since it was the only parameter that was found significantly associated with TBS after adjustment for all confounders. Nonetheless, we have now substituted it with a new model which maintains all covariates, as required by the reviewer, confirming that the effect of HDL-C on TBS is independent from age, menopause, BMI, waist circumference, physical activity and statins use, with Model’s R: 0.682 and R2: 0.465 (new table 3).

Variables

Unstandardized coefficient

Standardized coefficient

p value

β

Standard Error

β

Age

-0.004

0.005

-0.162

0.464

Menopausal state

0.008

0.005

0.303

0.175

BMI

-0.007

0.008

-0.304

0.411

Waist circumference

0.002

0.004

0.199

0.634

HDL

0.006

0.002

0.603

0.028

Physical activity

-0.056

0.069

-0.214

0.432

Statin treatment

0.016

0.061

0.066

0.793

Vitamin D

0.002

0.061

0.066

0.793

Constant

0.686

0.514

-

0.205

  1. Additional comment: The authors have not assessed trabecular “microarchitecture” directly and the title is misleading. TBS reflects “altered bone texture”. I think “microarchitecture” should be removed from the title, aims, results, discussion for the reasons stated below.

The term “microarchitecture” has been removed from the title, aims, results, discussion, as suggested by the reviewer, and substituted with the expressions “bone quality” or “texture”. The clinical role and value of the TBS assessment has been made now explicit in the Introduction: “The Trabecular Bone Score (TBS), is a BMD-independent parameter of the spine DXA image which evaluates bone texture and is, therefore, an indirect estimator of the trabecular bone quality” (Page 2 lines 71-73).

  1. Additional comment: This standardized beta coefficient does not appear to have been corrected.

We apologize for the refuse and have now corrected the coefficient’s sign (Page 5, line 180).

Reviewer 2 Report (New Reviewer)

The authors have improved their manuscript.

Author Response

The authors have improved their manuscript.

We thank the reviewer for the insightful comments made which allowed to improved our manuscript.

Round 3

Reviewer 1 Report (New Reviewer)

I have no further comments.

This manuscript is a resubmission of an earlier submission. The following is a list of the peer review reports and author responses from that submission.

Round 1

Reviewer 1 Report

1) Has the sample size been calculated? 

2) Why was randomization not used ?

3) What is the definition of the typology of the study design?

4) Why is there no Prism flow chart ?

5) What is the result of the physical activity intensity of each of the two groups?

6) Under which chemical form did vitamin D supplementation occur?  

7) Why result of physical activity intensity is not present in Table 1 and 3   ?

8) Statin treatment was identical as molecule for all patients with DMT2 ?

Author Response

  • Has the sample size been calculated? 

We thank for this point. The sample size has been calculated according to data from the cross-sectional study by Leslie WD. et al [20], reporting mean± SD TBS=1.245 ± 0.125 in non-diabetic women vs TBS= 1.194 ±0.112 in T2DM women, in BMI, age and disease- adjusted analysis (see also: reference [49]). Based on these findings, n= 116 individuals per group were needed to have sufficient statistical power to detect the effect of T2DM on TBS, with an a-error= 0.05 and a power of 90% (Rosner B. Fundamentals of Biostatistics. 7th ed. Boston, MA: Brooks/Cole; 2011). These calculation has been now reported in the Statistics section of the manuscript  (Page 9, lines 414-419).

  • Why was randomization not used?

This is an observational case-control study aimed at investigating possible clinical predictors of bone alterations in a population of women with T2DM compared to controls. Given the fact that we had two different cohorts to select, the procedure of randomization is not applicable (see also: Tenny S, Kerndt CC, Hoffman MR. Case Control Studies. In: StatPearls. Treasure Island (FL): StatPearls Publishing; March 28, 2022).

  • What is the definition of the typology of the study design?

This is an observational case-control study. We apologise if this was not clearly stated in the previous version of this manuscript and have now added this definition in the text (page 8, line 358).

  • Why is there no Prism flow chart ?

The Prisma flow chart is a type of flowchart used to report systematic reviews and meta-analyses, therefore its use was not suitable for this manuscript (original research, observational case-control study). Probably the reviewer was suggesting the STROBE checklist, that we have added as supplementary material.

  • What is the result of the physical activity intensity of each of the two groups?

Results from the International Physical Activity Questionnaire administrated to our study participants showed that 46% T2DM patients had a sedentary lifestyle (none-low physical activity), 46% practiced moderate physical activity and 8% were engaged in high intensity exercise programs. In women without T2DM we observed a prevalence of 46.1% individuals in the none-low intensity subgroup, 43.6% in the moderate intensity and 10.3% in the high intensity group; no significant difference was observed between T2DM and non-T2DM subjects. No association was found between TBS and physical activity in the study population. We apologise if these results were missing in the previous version of this manuscript and have added them now in Table 1 and Supplementary Table 1S (previous numbered as Table 3), as suggested by the reviewer (see also response to point 7).

  • Under which chemical form did vitamin D supplementation occur?  

The chemical form of vitamin D supplementation was oral Colecalcipherol (Vitamin D3) therapy, as prescribed by the general practitioner / endocrinologist. This information has been now reported in the manuscript: “Similar prevalence of vitamin D supplementation (consisting of oral colecalcipherol (Vitamin D3) treatment) was reported in women with or without T2DM” (Page 3, lines 177-179).

  • Why result of physical activity intensity is not present in Table 1 and 3   ?

We apologise for this refuse and have now added this information in the manuscript, as suggested by the reviewer (Table 1 and Supplementary Table 1S, previously numbered as Table 3; see also response to point 5).

  • Statin treatment was identical as molecule for all patients with DMT2?

For the purposes of this study, we considered the presence or not of statin therapy, as a categorical variable. In our analyses, statin treatment was not associated to study outcomes and did not represent a confounder in the relationship between impaired TBS and low HDL at the multivariate regression analysis. The relative prevalence of use of molecules belonging to the statin class in T2DM patients was: atorvastatin 37.7%, simvastatin 34% and rosuvastatin 28.3% and was comparable to the one reported in the control group (atorvastatin 40%, simvastatin 28% and rosuvastatin 32%). Similarly, to what observed when the overall statin use (as a dummy variable) was entered in the analysis, no relationship was found between any specific molecule belonging to the statin class and the TBS. We thank the reviewer for this comment and have now added this information in the new version of our manuscript: “Statin treatment was significantly more frequent in patients with T2DM (p=0.002); no difference was observed in the prevalence of treatment with molecules belonging to this class between T2DM patients and controls (atorvastatin 37.7%, simvastatin 34% and rosuvastatin 28.3% vs atorvastatin 40%, simvastatin 28% and rosuvastatin 32%, respectively)” (Page 3, lines 173-177).

Reviewer 2 Report

Dear authors, 

Thanks for your submission. The topic is interesting, but not well presented. There is across the board a formatting issue with spaces and words that are combined.  More important the way that stats are presented in inconsistent and confusing. All these make the reader to lose focus on the RQs.

Abstract doesnt agree with the title is not focused. Please amend accordingly. Results are very difficult to follow as even though there is a control group involved, in the majority of the stats the analysis was performed in the whole population. Keep the variables of interest that you introduce in the Intro, organize the flow so it is easy to understand and follow your logic.

Fix your stats, the flow and focus on the main variables of interest.

Author Response

Reviewer 2:

Dear authors, 

Thanks for your submission. The topic is interesting, but not well presented. There is across the board a formatting issue with spaces and words that are combined.  More important the way that stats are presented in inconsistent and confusing. All these make the reader to lose focus on the RQs. Abstract doesn’t agree with the title is not focused. Please amend accordingly. Results are very difficult to follow as even though there is a control group involved, in the majority of the stats the analysis was performed in the whole population. Keep the variables of interest that you introduce in the Intro, organize the flow so it is easy to understand and follow your logic. Fix your stats, the flow and focus on the main variables of interest.

We thank the reviewer for the important comments and issues raised in the revision process and have now deeply revised this manuscript focusing on the RQs, keeping the variables of interest and illustrating the statistics applied to demonstrate our research hypotheses. All points raised by the reviewer have been answered; the point by point list of responses is reported below.

  1. Abstract: I would rewrite this to focus on the main RQ that relates to the title - otherwise you need to change the title to match whatever you did. At this stage, abstract is not focused and covers a wide range of health and metabolic related factors.

We thank the reviewer for this important comment, and acknowledge that in the initial version, the abstract was not focused on RQ. Therefore, we have now rewritten the abstract, describing the RQs and the statistics used to demonstrate our study hypotheses; moreover, we have now focused on results obtained in T2DM women, as stated in the title (Page 1 lines 19-30).

  1. Add in the Methods something to indicate the Stats used so they match the RQ, and the Results.

A brief description of the Statistics used in this research, according to RQ and Results, has been now added in the Abstract: “DXA-derived indexes were correlated to metabolic parameters by bivariate analyses, and significant variables were entered in multivariate adjusted models to detect independent determinants of altered bone status in diabetes” (Page 1, lines 19-22).

  1. Use keywords other than the one that you use in the title so you can increase the chances to have the article "discovered" when people are searching online.

The keywords list has been changed as follows: “osteoporosis; osteopenia; fracture risk; lipid metabolism; metabolic syndrome; insulin resistance; trabecular bone score”, omitting words already used in the title, as suggested by the reviewer.

  1. Page 1 line 40: references to support the "over the years a considerable amount of data"

We thank the reviewer for this comment and have now added the reference [5], which looks at data gathered over the last years indicating different possible mechanisms that lead to bone fragility in diabetes (Page 1, line 44).

  1. Page 1, line 41: references to support the ambiguity

We have now added the review article at reference 5  (Murray, C.E.; Coleman, C.M. Impact of Diabetes Mellitus on Bone Health. Int J Mol Sci 2019, 20(19):4873. doi: 10.3390/ijms20194873) where the authors summarize the available literature on the impact of diabetes mellitus on bone health in vitro and in vivo concluding that,  although different pathogenic mechanisms have been proposed in the recent years, no conclusive model exists explaining the link between glucose metabolism impairment and bone fragility (Reference 5).

  1. Table 1: T2DM+: what is this? I am confused with the way that you present the sig and non-sig values. Be consistent with the literature.

The expression “T2DM+” has been substituted with “T2DM individuals”.

In this manuscript’s tables, we reported both statistically significant and non-significant p values. P values<0.05 were considered statistically significant and marked in bold. We have now specified better the threshold for statistical significance and explained the meaning of the asterisk and the bold use in the figure legend: “Student's T test applied; the asterisk (*) indicates that χ2 test was calculated. P values <0.05 were considered statistically significant with a confidence interval (C.I.) of 95%. Significant p values are reported in bold”(Table 1).

  1. Table 2: T2DM+: what is this? Why not an asterisk? Define again the T2DM.

The expression “T2DM+” has been substituted with T2DM individuals” also in this table.

In this table, the asterisk indicates that the c2 test was applied. The comparison between lumbar spine BMD in T2DM vs. non-diabetic individuals was calculated by Student’s T-test, as for continuous variable, and the p value (= 0.031) was reported in bold, indicating that this value is statistically significant. We have not specified in the Table legend that “Student's T test was applied for all comparisons. Significant p values are reported in bold”(Table 2). T2DM was defined again in the Table legend.

  1. Table 3: don’t follow the bold values with the asterisks and the non bold with the asterisks. it is confusing as normally we bold and place asterisks to sig values, but is here I see it all over the place, with no clear logic.

In the manuscript tables, the * is used to specify the type of test applied, when different tests are applied in the same table. Differently, the bold indicates statistically significant results. We apologize if it was somehow confusing and have now specified the logic behind data presentation in the Figure legend: “Pearson’s coefficient calculated; the asterisk (*) indicates that Spearman coefficient was applied, as appropriate. Significant p values are reported in bold”(Supplementary Table 1S, previously numbered as Table 3).

  1. Figure 1. what is the T2DM+ and -? / define these

T2DM+ and – has been substituted with “T2DM individuals” and “controls”. T2DM has been defined in the legend: “Figure 1. Trabecular Bone Score categories distribution in patients with T2DM (type 2 diabetes mellitus) and healthy controls”(Figure 1).

  1. Table 4. why not bold and asterisks here?

We have explained more clearly the meaning of asterisk and bold in the tables legend (see response to points 6, 7 and 8). In order to make easier to understand and follow the manuscript flow, we have now re-organized the article and focused on the main variables of interest, as suggested by the reviewer. Table 4 did not add major information and has been therefore deleted from the revised version of this manuscript.

  1. Table 5. Define the values presented in the table.

We have now defined the values presented in the table in the figure legend, as indicated by the reviewer: “Pearson’s coefficient calculated; the asterisk (*) indicates that Spearman coefficient was applied, as appropriate. Significant p values are reported in bold.

T2DM: Type 2 Diabetes Mellitus; BMI: Body Mass Index; FBG: Fasting Blood Glucose; HbA1c: Glycosylated Hemoglobin; FBI: Fasting Blood Insulin; HOMA-IR: Homeostatic  Model Assess-ment for Insulin Resistance; HDL: High-Density Lipoprotein; LDL: Low-Density Lipoprotein; AST: Aspartate Aminotransferase; ALT: alanine aminotransferase; GGT: Gamma-Glutamyl Transferase; TSH: Thyroid Stimulating Hormone; PTH: Parathyroid Hormone”(Table 3, previously numbered as Table 5).

  1. Table 6. TBS is the dependent variable: Why grouped them all and not use the cut-off points?

In the multivariate regression analysis, the TBS was considered as a continuous variable since the distribution of patients within the TBS classes was markedly unbalanced. In particular, only 5% of T2DM patients had normal TBS according to the standard cut-off. This did not allow to use this category as a comparator either in a logistic regression model using TBS as a categorical variable or in ordinal model. Therefore, we built a multivariate linear regression model which demonstrated that reduced TBS, as a continuous value, associated with lower HDL in a dose-dependent manner and regardless of metabolic covariates in women with T2DM.

  1. For the first time, to our knowledge: I would delete this.

This sentence has been deleted as suggested by the reviewer.

  1. Previous reports investigating the association between lipid profile and bone metabolism in terms of BMD and fracture risk have shown controversial results, mostly due to the heterogeneity of the studies”: what is the factor that makes this study more homogeneous compared to others?

With this affirmation, we meant that differences in the findings on the association between lipid profile and bone metabolism reported in previous studies, might be likely attributable to differences in study design, populations and experimental setting/methods among these investigations. We acknowledge that this expression could have been confusing and have now re-formulated it: “Previous reports investigating the association between lipid profile and bone metabolism in terms of BMD and fracture risk have shown controversial results, likely due to differences in study design, populations and methodology applied” (Page 6, lines 262-263).

  1. “To our knowledge, there are no studies that have investigated this association in relation to glucose tolerance profile and T2DM”: if this is the goal, please focus on this and present stats dedicated to glucose profile to make your point stronger.

In this study, we investigated bone architecture in relation to the presence of T2DM and metabolic determinants of altered bone mass and quality in diabetic patients. In our statistics, indeed, we investigated the relationship between indicators of bone mass density / architecture and glucose profile, as evaluated by fasting blood glucose, HbA1c, fasting blood insulin and insulin resistance. We have now formulated this sentence more clearly in order to provide a more accurate description of our study goal: “To our knowledge, there are no studies that have investigated the association between qualitative bone and metabolic profile in T2DM” (Page 6, lines 268, page 7, lines 279-280).

  1. “Finally, growing evidence is suggesting that also inflammation plays a fundamental role in the development of osteoporosis [37]. HDL may indirectly influence bone metabolism also for its anti-inflammatory effects on bone microenvironment [38]” Please elaborate more on this.

The potential role of HDL in influencing bone microenvironment has been now discussed more in detailed, as suggested by the reviewer: “Finally, growing evidence is suggesting that also inflammation plays a fundamental role in the development of osteoporosis. Data indicate that pro-inflammatory cytokines such as IFN-γ, IL-17A, IL-15 and TNFα promote osteoclastogenesis and impair osteoblastogenesis [38]. Interestingly, HDL can inhibit the interaction between T lymphocytes and antigen presenting cells (APCs), preventing the activation of the latter and associated TNFα and IL-1β production [39], suppressing gene expression of mediators of the type I interferon response pathway [40]. Therefore, HDL may indirectly influence bone metabolism for its systemic and tissue-specific anti-inflammatory properties [41]” (page 7, lines 305-312).

  1. “The association between glycaemic control and bone fragility is controversial. It is known that some antidiabetic drugs affect bone metabolism. In order to identify non-glycaemic predictors of altered bone metabolism, our study was designed to minimize the influence of glycaemic control and different anti-diabetic therapies on bone health, selecting patients in good glycemic control (HbA1c≤53 mmol/mol) and treated only with metformin, without history of other antidiabetic treatments”: Citations here?

Citations [25-28] have been entered here, as suggested by the reviewer (Page 7,  line 314).

  1. “As a reference group, we also recruited 117 healthy women with age and BMI distributioncomparable to the T2DM cohort (mean age 61.91± 5.8 years, BMI 32.64±7.12 234 kg/cm2)”: we cannot have one sentence a paragraph - we need at least 2 sentences to constitute a paragraph.

This sentence has been moved up and entered in the paragraph on study population description (Page 8, lines 364-366).

  1. Present here more clearly the stats that were used to answer the RQs- Keep it neat and clear.

We thank the reviewer for this comment and have now explained better the statistical procedures used to answer the RQs of this manuscript. Moreover, the procedures for sample size and power calculation have been now described in the Statistics section: “Sample size and power calculation. The sample size of this investigation has been calculated according to data from the cross-sectional study by Leslie WD. et al [20] finding mean ±SD TBS = 1.245 ±0.125 in non-diabetic women vs TBS= 1.194 ± 0.112 in T2DM women, in BMI, age and disease- adjusted analysis (see also [49]. Based on these findings, n= 116 individuals per group were needed to have sufficient statistical power to detect the effect of T2DM on TBS, with an a-error= 0.05 and power= 90%”; “Finally, a the multivariate linear regression model was built to identify parameters independently associated with TBS value in T2DM women, entering age, menopausal status, variables associated with TBS at the bivariate analyses and potential confounders of HDL levels (i.e. statin treatment, physical activity)” (Paragraph 4.4, page 9, lines 414-419 and 428-431).

Round 2

Reviewer 2 Report

Dear Authors,

Thanks for the resubmission. Sadly this effort suffers again from the same issues. First your formatting is off, spaces again between words/numbers are missing. This is a sloppy submission that creates bias. 

Most important though is the way that your run and presented your Stats. Ask help from a statistician as your results are incorrect, therefore, the Discussion and Conclusion sections are misleading. Tables are misleading, incorrect presentation of categorical variables, correlations are not presented like this, regression tables miss important info and no where to examine the different models. 

Methods are sketchy. Write again VERY CAREFULLY the manuscript, ask help from a Statistician. These 101 Stats class - use as guide the https://academic.oup.com/jcem/article/98/2/602/2833108

I will advise for one more attempt, but this new should be stellar. I dont want to highlight again the same issues.

Author Response

Point-by point response to Reviewer 2.

Dear Authors,

Thanks for the resubmission. Sadly this effort suffers again from the same issues. First your formatting is off, spaces again between words/numbers are missing. This is a sloppy submission that creates bias. Most important though is the way that your run and presented your Stats. Ask help from a statistician as your results are incorrect, therefore, the Discussion and Conclusion sections are misleading. Tables are misleading, incorrect presentation of categorical variables, correlations are not presented like this, regression tables miss important info and no where to examine the different models. Methods are sketchy. Write again VERY CAREFULLY the manuscript, ask help from a Statistician. These 101 Stats class - use as guide the https://academic.oup.com/jcem/article/98/2/602/2833108.

I will advise for one more attempt, but this new should be stellar. I don’t want to highlight again the same issues.

We have now revised and changed the statistics (including the presentation) as suggested, performed additional test as requested. The results are solid and confirmed, and therefore the discussion is supported by the results.

Abstract. Differences in bone alterations between 2 groups : T2DM and Controls.

We have now re-formulated this sentence as suggested by the reviewer: “Our study aimed to explore differences in bone alterations between T2DM women and controls and to assess clinical predictors of bone impairment in T2DM” (Page 1, lines 14-16).

Good sample, needs just a justification on how these numbers came up.

We have added further information on the procedures applied for sample size calculation in the Statistics section (Page 9, lines 330-335)

what metabolic parameters? need to define her even in the abstract.

This has been now specified in the abstract: “DXA-derived indexes were correlated to metabolic parameters, such as glycemic control and lipid profile” (Page 1, line 21).

Need to define HDL too.

HDL has been defined also in the abstract: “TBS directly correlated with plasma high-density lipoprotein (HDL) [… ] levels” (Page 1, line 27).

Nice, if there is a need for another revision please provide a clean version, this is one with track change is difficult to read.

We apologize if the previous version with track changes was difficult to read and have now provided a version with changes highlighted in yellow. However, we were advised that “Any revisions to the manuscript should be marked up using the “Track Changes” function….” by the editorial office.

Table 1. These are differences, so revise the title.

The title of Table 1 has been changed according to the reviewer’s comment: “Table 1. Differences in clinical and biochemical features between T2DM patients and controls” (Page 2, line 90).

what are these mean and SD, SE? what???

In tables, data are shown as mean values ± standard deviation (SD) or percentages. This information, reported in the statistics section, has now been added under each table (Page 3, line 92).

This doesn’t make sense- I cannot tell what the difference is ...I see T2DM was 92 and Control was 97. is the difference on the yes or no? who had the me menopause status? Moreover the did not differ in age, so the younger one had 97 and the older T2DM had 92 but what??? You need to fix this.

In this table, we reported that 92.1% T2DM and 97.4% controls were in postmenopausal state at the time of study enrollment. This difference is not statistically significant (p= 0.063). We apologize if the presentation of this information in the table was not clear and have now amended it. The asterisk, which indicated the test applied (chi-squared) has been deleted since it could be misleading, as observed by the reviewer (Table 1, page 3).

Same here, these % make no sense on what is the reference - delete the asterisk.

Asterisks have been deleted from the tables.

No need for this - we will read the tests in the Stats section. We assume it is T-test, independent groups.

The description of the test used, already reported in the Stats section, has been deleted from the Tables’ legend, as suggested by the reviewer.

This is Chi square test of association for independent samples - no need for the asterisk- all results are calculated - keep it simple.  bold sig values for every test. In the stats section describe how you tested differences between categorical-nominal values and continuous.

We have amended all the Tables, deleting the asterisks and the description of the tests used; significant values are reported in bold for every test. Tests used are reported in the Stats section (Page 9, lines 310-313).

“P values <0.05 were considered statistically significant with a confidence interval (C.I.) of 95%” if you have this, then the table needs to have the CI reported for each value.

This sentence has been deleted from the legend.

“As expected” Delete.

The expression “as expected” has been deleted.

“Statin treatment was significantly more frequent in patients with T2DM (p=0.002); no difference was observed in the prevalence of treatment with molecules belonging to this class between T2DM patients and controls (atorvastatin 37.7%, simvastatin 34% and rosuvastatin 28.3% vs atorvastatin 40%, simvastatin 28% and rosuvastatin 32%, respectively)”:

???? dont follow..what molecules in specific? present these types of statin in the table and indicate what is the reference criterion.

This sentence has been now reformulated more clearly, and the type of statin used is now indicated in Table 1, as suggested by the reviewer (Table 1, page 3).

Table 2: These are differences so revise the title - here you may add the T-test.

We have revised the title of table 1, according to the reviewer’s comment: “Differences in bone parameters between patients with T2DM and controls” (Table 1, page 2).

“Individuals” Delete.

We deleted the term “individuals” from all tables.

1.010 ± 0.165: what are these?

These are mean values ± standard deviation (SD). This information is now reported in the Tables legend (Page 4, line 118). We apologize if the data presentation was not clear in the previous version of this manuscript.

“Student's T test applied for all comparisons”: same as earlier- this will be explained.

The description of the test used has been now deleted from the Tables’ legend, as suggested by the reviewer. Tests used are reported in the Stats section (Page 9, lines 310-313).

T2DM patients were more represented in the degraded (TBS≤1.2) and partially degraded (1.2<TBS≤1.35) bone category (52% vs 48%, 43% vs 40% respectively) and 131 less represented in the normal bone category (TBS>1.35; 5% vs 12%, p=0.04) compared to 132 controls: put this in the table.

These data have been now included in Table 2, as suggested by the reviewer (Table 2, page 4).

Figure 1. use bar graphs not a pie chart - we don’t care about representation. we care about differences in the sub groups. are these sig ??? only the normal? difficult to infer- use either a table of bar graphs.

This looks like a Simpson paradox - discuss it in Discussion.

In this study, we found that the prevalence of individuals with normal bone structure is lower in the T2DM group vs controls (5% vs 12%, p= 0.041). For this analysis, we used a chi-squared test entering, as categorical variables, presence or not of T2DM and “normal” vs “partially degraded + degraded bone” structure. We have now created a new bar graph as suggested by the reviewer, re-formulated the sentence in the results and provided a more clear explanation of the Stats procedure applied in the Statistics section. Moreover, to further investigate the association between diabetes and impaired TBS measurement, we have now performed new analyses based on previous investigations on this topic (https://academic.oup.com/jcem/article/98/2/602/2833108), as indicated by the reviewer. We stratified the whole study population in tertiles according to TBS value and BMD measurements (total hip, femoral neck and lumbar spine) and calculated odds ratios for the lowest vs highest tertile with 95% confidence intervals (CIs) associated with diabetes status, by logistic regression analyses adjusted for age, menopausal status, and BMI. We found that the adjusted OR for TBS in the lowest tertile (vs highest tertile reference) associated with diabetes was significantly more than 1 (OR: 2.47 (CI95%: 1.19-5.16), p= 0.016). Conversely, diabetes status was confirmed to be associated with greater lumbar spine BMD. No significant ORs were obtained for BMD measured in the other district, confirming what already shown in the T test comparison between T2DM and controls (Table 2, page 4). These new analyses have been described in the Statistics section, described in the Results and shown in a new figure (Figure 2 page 5).

“When dividing the study population according to validated TBS cut-offs [29], we observed that the prevalence of normal bone structure was significantly lower in the T2DM groups compared to controls (p<0.041)” (Results, page 4, lines 123-125, new Figure 1).

“Differences between independent groups were compared by Student’s T test for continuous variables and by χ2 test for categorical variables” (Statistics, page 9, lines 310-313).

 “To test the association between presence of diabetes and DXA-derived indexes of bone health, we stratified the whole study population in tertiles according to TBS value and BMD measurements (total hip, femoral neck and lumbar spine) and calculated odds ratios for the lowest vs highest tertile with 95% confidence intervals (CIs) associated with diabetes status, by logistic regression analyses adjusted for age, menopausal status, and BMI”(Page 9, lines 314-319).

“The presence of T2DM was associated with TBS in the lowest tertile vs highest tertile with adjusted OR: 2.47 (C.I. 95%: 1.19-5.16, p= 0.016). Conversely, diabetes status was con-firmed to correlate with greater lumbar spine BMD (OR: 0.43, C.I. 95%: 0.21-0.89, p= 0.024), but not with BMD values measured in the other districts (femoral neck OR: 0.65, C.I. 95%: 0.22-1.92, p= 0.433; total hip BMD OR: 0.56, C.I. 95%: 0.28-1.12) (Figure 2) (Results, page 5, lines 146-151).

Figure 1. Comparison between prevalence of normal vs altered bone structure in patients with type 2 diabetes mellitus (T2DM) and healthy controls.

Figure 2. Odds ratios (95% C.I.) for T2DM patients to be in the lowest tertile of TBS and BMD measurements, adjusted for BMI, age and menopausal status.

Finally, we have commented the presence of this Simpson’s paradox in the Discussion section, according to this reviewer’s suggestion: “Although the mean TBS value was similar in T2DM and controls, when dividing the two cohorts in subgroups according to TBS cut-offs, we observed that the prevalence of individuals with normal bone structure was less than an half in the T2DM group than in controls, also in presence of normal or even increased BMD, in line with other studies [20, 44]. Similarly, we found that the presence of T2DM was associated with TBS values in the lowest tertile with highest tertile as a reference, with OR 2.47 (C.I. 95%: 1.19-5.16) adjusted for age, menopausal status, and BMI” (Discussion, page 7 lines 252-255; page 8, lines 256-258).

Why correlates and not going straight to regression and create your models? wrong way to present it- Use a correlation matrix. age correlates with what?

Following this reviewer’s indications, we re-analyzed our data e presented the statistics showing directly the univariate regressions and the multivariate models, that now include R and R2 values for each one (Results, Table 3, page 6). We have re-written the entire Statistics section to describe step by step the statistical procedures applied and entering all the changes suggested by the reviewer, following the suggested reference from JCEM:

“Descriptive statistics are presented as mean ± standard deviation (SD) for continuous variables or percentage for categorical variables. Differences between independent groups were compared by Student’s T test for continuous variables and by χ2 test for categorical variables. In order to test the association between presence of diabetes and DXA-derived indexes of bone health, we stratified the whole study population in tertiles according to TBS value and BMD measurements (total hip, femoral neck and lumbar spine) and calculated odds ratios for the lowest vs highest tertile with 95% confidence intervals (CIs) associated with diabetes status, by logistic regression analyses adjusted for age, menopausal status, and BMI. Moreover, the prevalence of individuals with normal vs degraded bone structure within T2DM group and controls was calculated by χ2 test. In order to identify predictors of impaired TBS in T2DM, TBS was analyzed as the de-pendent variable in univariate and multivariate regression models that included age, menopausal status and metabolic parameters. Backward regression analyses were created from a saturated full model gradually eliminating variables to find a reduced model that best explained the data, i.e. changes of TBS in relation to entered parameters. R2 was calculated as a goodness-of-fit measure for our regression models; both R and R2 were re-ported for each model. A p value < 0.05 was taken to indicate a statistically significant effect. Statistical analyses were performed with SPSS for MacOs (version 27, SPSS Inc, Chicago, Illinois)” (page 9, lines 310-329).

Table 4. We need R and R2 - table is incomplete. Where are your models?

R and R2 value for the final model previously shown in Table 4 are: R= 0.68, R2= 0.47. We have now shown all the models built, with R and R2 values, according to this reviewer’s comment (new Table 3, page 6).

“In this study we evaluated bone health in a population of women with diabetes compared to a control group and investigated possible clinical and biochemical predictors of bone quality in conditions of diabetes.” Discussion. These need to be re-written with accuracy and extra caution to reflect the results presented in the tables. still it is not clear to me - what the control was. you need in the methods to categorize what variables are the clinical and what are the biochemical. Question ...HDL is clinical or biochemical? or Both?  “We demonstrated that women with T2DM have more degraded bone microarchitecture as assessed by TBS compared to healthy age- and BMI- comparable women, also in presence of normal or even increased BMD”:  is that right? or only for a specific cut off point? Table 2 says no difference.

These paragraphs have been re-formulated according to the reviewer’s comment:

“In this study, we evaluated bone health in a population of T2DM women compared to women without diabetes, comparable for age and BMI, recruited as a control group and then we explored metabolic predictors of bone quality in T2DM. We demonstrated that women with T2DM have more degraded bone microarchitecture, as assessed by TBS compared to healthy women, also in presence of similar or even increased BMD” (page 6, lines 190-194).

“Although the mean TBS value was similar in T2DM and controls, when dividing the two cohorts in subgroups according to TBS cut-offs, we observed that the prevalence of individuals with normal bone structure was less than an half in the T2DM group than in controls, also in presence of normal or even increased BMD, in line with other studies [20, 44]. Similarly, we found that the presence of T2DM was associated with TBS values in the lowest tertile, with highest tertile as a reference, with OR 2.47 (C.I. 95%: 1.19-5.16) adjusted for age, menopausal status, and BMI” (Discussion, page 7 lines 252-255; page 8, lines 256-258).

What do you mean with consecutive???

For consecutive patients we mean all eligible patients identified during the enrollment period, among those who attended our clinics (i.e. metabolic evaluation for diabetes in outpatient clinics) (Hagopian LP. The consecutive controlled case series: Design, data-analytics, and reporting methods supporting the study of generality. J Appl Behav Anal. 2020 Apr;53(2):596-619. doi: 10.1002/jaba.691). We thank the reviewer for this comment and have now described better this recruitment procedure in the Methods: “[…] we recruited 126 eligible women with T2DM consecutively referring to our Endocrinology and Diabetes outpatient clinic of Sapienza University, Rome, Italy, for metabolic evaluations” (page 8, lines 266-268).

BMI 31.63 ± 7.09 kg/m2: have these in Table 1 already.

This information has been deleted from the text.

What is a metabolic evaluation?

For metabolic evaluation, we mean all the procedures carried out for evaluating the presence and severity of diseases affecting the metabolism, such as obesity, impaired glucose regulation and diabetes mellitus, systemic blood hypertension, dyslipidemia… These procedures are all described in the section “4.2. Clinical and biochemical evaluations”. We apologize if this point was not clear and have now re-named the paragraph 4.2 “Metabolic evaluations” (page 8, line 280).

BMI???? UNITS?????????

We apologize and have now corrected this refuse: BMI: 32.64±7.12 kg/m2 (page 8, line 274).

“Based on these findings, n= 116 individuals per group were needed to have sufficient statistical power to detect the effect of T2DM on TBS, with an a-error= 0.05 and power= 90%”. Based on the Mean and SD I calculated 115 per group with effect size 0.42, same power and alpha. = provide more detail how did you do the sample size calculation - software used.

Here you find the detailed sample size calculation, according to Rosner B. Fundamentals of Biostatistics. 7th ed. Boston, MA: Brooks/Cole; 2011 (https://clincalc.com/stats/samplesize.aspx).

not the right way to cite the SPSS.

We have now re-written this sentence: “Statistical analyses were performed with SPSS for MacOs (version 27, SPSS Inc, Chicago, Illinois)” (page 9, lines 328-329).

”mean ± standard deviation (SD) and categorical variables as percentages”: tables need that too. tables need to be able to stand alone, without any text info.

This has been added to all the Tables, as suggested by the reviewer.

“Univariate regression analyses were performed to test the association between categorical and continuous variables” why? Relevance to RQ?

The Statistics section has been fully revised by a statistician, providing better description of the Stats applied for the RQ. This sentence has been deleted from the new version of this manuscript.

“Multivariate regression age-forced models were built entering parameters significantly associated at the univariate/bivariate analyses. Finally, a multivariate linear regression model was built to identify parameters independently associated with TBS value in T2DM women, entering age, menopausal status, variables associated with TBS at the bivariate analyses and potential confounders of HDL levels (i.e. statin treatment, physical activity)”: This is stepwise, entered, what exactly and why? Provide references to justify the selection of the model. How did you evaluate the modes? AIC?

For this RQ, we have applied backward regression analyses that from a saturated full model gradually eliminates variables from the regression model to find a reduced model that best explains the data, i.e. changes of TBS in relation to entered parameters. Adjusted non-standardized and standardized beta from HDL in T2DM patients are shown in the new version of Table 3, along with R and R2 of each model. In the Supplementary data, we also provide the coefficients for the variables entered in each model.

Table 3. Multivariate linear regression analyses for reduced TBS from HDL values.

Models

HDL

R of the model

Rof the model

Non standardized Beta

Standardized Beta

p value

Model 1

0.006

0.536

0.003

0.686

0.470

Model 2

0.005

0.509

0.006

0.683

0.466

Model 3

0.005

0.507

0.006

0.680

0.463

Model 4

0.005

0.509

0.007

0.676

0.458

Model 5

0.005

0.495

0.012

0.659

0.435

Model 6

0.005

0.468

0.029

0.651

0.424

Models were adjusted for: Model 1. Menopausal status; Model 2. Menopausal status, BMI; Model 3. Menopausal status, BMI, Vitamin D; Model 4. Age, Menopausal status, BMI, Vitamin D; Model 5. Age; Menopausal status, BMI, Vitamin D, Physical aactivity; Model 6. Age, BMI, Menopausal status, Statin treatment, Vitamin D, Physical activity.

Supplementary Table 3. Multivariate linear regression analyses for reduced TBS from HDL values. Coefficients of variables entered in the models.

Models

Non standardized coefficients

Standardized coefficients

P value

Beta

Beta

Model 1

Constant

1.068

Menopausal status

-0.206

-0.446

0.01

HDL

0.006

0.536

0.003

 Model 2

Constant

1.154

BMI

-0.003

-0.106

0.528

Menopausal status

-0.202

-0.436

0.014

HDL

0.005

0.509

0.006

 Model 3

Constant

1.286

BMI

-0.004

-0.172

0.351

Menopausal status

-0.256

-0.553

0.014

HDL

0.005

0.507

0.006

Vitamin D

-0.001

-0.204

0.359

Model 4

Constant

1.36

Age

-0.002

-0.1

0.672

BMI

-0.004

-0.161

0.397

Menopausal status

-0.218

-0.47

0.113

HDL

0.005

0.509

0.007

Vitamin D

-0.001

-0.177

0.45

Model 5

Constant

1.35

Physical activity

0.016

0.063

0.724

Age

-0.002

-0.097

0.685

BMI

-0.004

-0.15

0.445

Menopausal status

-0.215

-0.465

0.126

HDL

0.005

0.495

0.012

Vitamin D

-0.001

-0.178

0.46

Model 6

Constant

1.391

Physical activity

0.021

0.08

0.674

Age

-0.002

-0.121

0.634

BMI

-0.004

-0.152

0.451

Menopausal status

-0.196

-0.423

0.198

HDL

0.005

0.468

0.029

Statin treatment

-0.02

-0.08

0.712

Vitamin D

-0.001

-0.192

0.441

Round 3

Reviewer 2 Report

Dear authors, thanks for the 3rd revised version. 

Starting from the obvious, I still was able to find formatting issues, spaces needed, paragraphs not indented, 5-6 lines sentences without a period, units not reported in any table, sig values not reported as bold in all tables, legends missing in tables so they cannot stand alone without the context.

Moving on to the most serious stuff. I am still bedazzled about the inconsistencies in the title, abstract and Intro. For example title points to HDL-C (which incorrect you have it as HDL) but your Intro has not reference to lipids or HDL-C. Abstract includes DVs that are not mentioned in the Intro and no connection between them is presented. As I stated in my previous reviews, this is a dataset that can produce more than 2 manuscripts, but the selection of the DVs needs to make sense. In your case it cannot, and you cannot cover everything as there are word limitations that need to meet.

Abstract remained chaotic, with many DVs reported that do not match the title and the purpose. Methods besides some minor things is okay. Stats still I dont see the OR and what new input this brings in. Your regression in the Stats is stated that is backward, from saturated model, but in the Results actually the model is forward, as you build up the model. Moreover, both methods are incorrect, because there is no HDL-C present, no clear indication of what in specific each modeling is comprised of, and in addition you dont tell us which model is the best and if there is a difference between your models. 

Your inclusion criteria had people taking metformin, vitamin D and calcium. But in your Intro and discussion you dont mention these at all. There are known effects of metformin on BMD, risk fractures, and lipid metabolism, as well as for vitamin D and calcium that you failed to discuss in the intro or control in the analysis, only vitamin D and statins were controlled.

In your discussion there is a complete absence of these issues, how they interact with each other and also I didnt see any limitations listed.

More in my detailed review of 88 comments.
